# Neutrophil/lymphocyte ratio and other blood cell component counts are not associated with the development of postmolar gestational trophoblastic neoplasia

**Antonio Braga**[1,2,3,4]*, **Ana Clara Canelas**[1], **Berenice Torres**[1], **Izildinha Maesta**[5],
**Luana Giongo Pedrotti**[6], **Marina Bessel**[6], **Ana Paula Vieira dos Santos Esteves**[1],
**Joffre Amim Junior**[1], **Jorge Rezende Filho**[1], **Kevin M. Elias**[7], **Neil S. Horowitz**[7], **Ross S. Berkowitz**[7]

1 Department of Obstetrics and Gynecology, Postgraduate Program in Perinatal Health, Faculty of Medicine, Maternity School of Rio de Janeiro Federal University, Rio de Janeiro, Rio de Janeiro, Brazil, 2 Department of Maternal Child Health, Postgraduate Program in Medical Sciences, Faculty of Medicine of Fluminense Federal University, Niterói, Rio de Janeiro, Brazil, 3 National Academy of Medicine, Young Leadership Physicians Program, Rio de Janeiro, Rio de Janeiro, Brazil, 4 Postgraduate Program in Applied Health Sciences, Vassouras University, Rio de Janeiro, Rio de Janeiro, Brazil, 5 Department of Gynecology and Obstetrics, Botucatu Trophoblastic Disease Center of the Clinical Hospital of Botucatu Medical School, São Paulo State University - UNESP, Botucatu, São Paulo, Brazil, 6 Hospital Moinhos de Vento, Porto Alegre, Rio Grande do Sul, Brazil, 7 Department of Obstetrics, Gynecology and Reproductive Biology, New England Trophoblastic Disease Center, Division of Gynecologic Oncology, Brigham and Women's Hospital, Harvard Medical School, Boston, Massachusetts, United States of America

* antonio.braga@ufrj.br

**Data Availability Statement:** Due to ethical issues related to the public disclosure of a database of a

## Abstract

### Objective

To relate preevacuation platelet count and leukogram findings, especially neutrophil/lymphocyte ratios (NLR) and platelet/lymphocyte ratios with the occurrence of gestational trophoblastic neoplasia (GTN) after complete hydatidiform mole (CHM) among Brazilian women.

### Methods

Retrospective cohort study of patients with CHM followed at Rio de Janeiro Federal University, from January/2015-December/2020. Before molar evacuation, all patients underwent a medical evaluation, complete blood count and hCG measurement, in addition to other routine preoperative tests. The primary outcome was the occurrence of postmolar GTN.

### Results

From 827 cases of CHM treated initially at the Reference Center, 696 (84.15%) had spontaneous remission and 131 (15.85%) developed postmolar GTN. Using optimal cut-offs from receiver operating characteristic curves and multivariable logistic regression adjusted for the possible confounding variables of age and preevacuation hCG level (already known to be associated with the development of GTN) we found that ≥2 medical complications at

disease that is not highly prevalent, coming from a delimited geographical area, with the potential to breach data confidentiality, even in the face of data de-identification, any data referring to the study "Neutrophil/lymphocyte ratio and other blood cell component counts are not associated with the development of postmolar gestational trophoblastic neoplasia" must be requested directly from the Research Ethics Committee of the Maternity School of Rio de Janeiro Federal University, with due justification analyzed by this institutional review board, in order to preserve the interests of research participants, following compliance with research standards in Brazil (CNS/CONEP/CEP 466/2012). The Research Ethics Committee of the Maternity School of Rio de Janeiro Federal University can be contacted directly through the institutional email, cep@me.ufrj.br, or directly by phone, 55.21.2285-7935.

**Funding:** This research was supported by the National Council for Scientific and Technological Development – CNPq (AB: 311862/2020-9), Carlos Chagas Filho Foundation for Research Support of the State of Rio de Janeiro – FAPERJ (AB: E-26/201.166/2022), Donald P. Goldstein MD Trophoblastic Tumor Registry Endowment (KME, NH, RSB), the Dyett Family Trophoblastic Disease Research and Registry Endowment (KME, NH, RSB) and Keith Higgins and the Andrea S. Higgins Research Fund (KME, NH, RSB). The funders had no role in study design, data collection and analysis, decision to publish, or preparation of the manuscript.

**Competing interests:** The authors have declared that no competing interests exist.

presentation (aOR: 1.96, CI 95%: 1.29–2.98, p<0.001) and preevacuation hCG ≥100,000 IU/L (aOR: 2.16, CI 95%: 1.32–3.52, p<0.001) were significantly associated with postmolar GTN after CHM. However, no blood count profile findings were able to predict progression from CHM to GTN.

## Conclusion

Although blood count is a widely available test, being a low-cost test and mandatory before molar evacuation, and prognostic for outcome in other neoplasms, its findings were not able to predict the occurrence of GTN after CHM. In contrast, the occurrence of medical complications at presentation and higher preevacuation hCG levels were significantly associated with postmolar GTN and may be useful to guide individualized clinical decisions in postmolar follow-up and treatment of these patients.

## Introduction

Hydatidiform mole (HM) is the most common form of gestational trophoblastic disease (GTD) and represents its benign spectrum [1]. It results from an abnormal fertilization, which presents as either of two clinical forms, complete (CHM) and partial hydatidiform mole (PHM), which differ from each other by their cytogenetic, histological, clinical and prognostic profile [2].

PHM originates from a diandric triploidy, with rare atypical villi, and a generally mild clinical presentation, which can progress to postmolar gestational trophoblastic neoplasia (GTN) in about 1–5% of cases. CHM, on the other hand, results from a diandric diploidy, exhibiting marked trophoblastic hyperplasia, exuberant clinical presentation and progression to postmolar GTN in about 15–20% of cases [1–4].

GTN is highly curable, even in multimetastatic disease, due to multiagent chemotherapy that, despite early and late toxicity, promotes disease remission [5, 6]. Patients with CHM who are carefully followed with human chorionic gonadotropin (hCG) monitoring are generally diagnosed with low-risk GTN and achieve high remission rates with single agent chemotherapy with low morbidity [7, 8].

The prediction of postmolar GTN has made only limited progress. The prior established prognostic parameters have failed to be highly predictive of the occurrence of postmolar GTN, such as the presence of medical complications and the trophoblast pathology [9]. Promising methods such as evaluation of ploidy, markers of cell proliferation and apoptosis [10, 11], oncogene expression [12] and analysis of circulating micro ribonucleic acid (RNA) [13], are complex and often unavailable in clinical practice, especially in developing countries, where there is a higher incidence of HM. Knowing which patients are at high risk of developing postmolar GTN and employing strict postmolar follow-up may facilitate early diagnosis of GTN. Furthermore, identifying patients with minimal probability of developing postmolar GTN may allow shorter hCG follow-up, particularly after hCG normalization, with a consequent decrease in patients lost to follow-up, patients' anxiety and the costs of prolonged hormonal surveillance.

Neutrophil/lymphocyte ratios (NLR) and platelet/lymphocyte ratios (PLR) have been associated with prognosis for many different non-malignant and malignant diseases, including gynecological tumors, suggesting these hematological findings of inflammation and oxidative

stress in peripheral blood may be involved in disease pathogenesis and may serve as a prognostic marker as well [14, 15]. However, the studies are heterogeneous and present controversial results. Moreover, the data for the assessment of these parameters in the prognosis of HM are scarce and come from series with few cases evaluated [16–18].

This study evaluates the potential relationship between the blood count results, especially the NLR, and the occurrence of postmolar GTN among Brazilian women. Our results are particularly important because of the need for simple prognostic variables that might be obtained from complete blood cell counts that are systematically requested before molar evacuation and are widely available in clinical practice, even in countries with limited health resources.

## Material and methods

### Study design

This is a retrospective historical cohorts study of patients with HM followed at the Rio de Janeiro Trophoblastic Disease Center—Maternity School of Rio de Janeiro Federal University (Rio de Janeiro—RJ, Brazil, data entered by ACC and BT and audited by AB), from January 1st 2015 to December 31st 2020. This study was approved by the local Institutional Review Board associated with the Brazilian Research Ethics Committee of the Maternity School of the Rio de Janeiro Federal University (CAAE 49462315.0.0000.5275, opinion number 1.244.337 of September 25th, 2015 and 3.972.252 of April 15th, 2020 –amendment). The study was done with anonymized patient records, so the Ethics Committees waived the need for obtaining individual informed consent.

### Study participants

The participants in this study were women who underwent molar evacuation at the Rio de Janeiro GTD Reference Center (CR) [19] and diagnosed with CHM, confirmed by histopathology [20]. We decided to evaluate only the cases of CHM because, in addition to being epidemiologically more frequent in our sample more reliably diagnosed than PHM, CHM have a higher occurrence of postmolar GTN, providing a more appropriate correlation with the primary outcome of this study.

All patients included in this study were followed to six months after hCG normalization or until the diagnosis of postmolar GTN. Hormonal contraception was advised during all follow-up [21]. Patients with histopathological diagnosis of PHM or twin molar pregnancy, those who became pregnant or were lost to follow-up were excluded from this study.

To obtain the complete blood count, no fasting was required. In all patients, the blood test was collected within 6 hours before surgery. The blood collection area was sanitized with cotton and alcohol. An elastic band was attached above the area to be punctured, so that the vein could be clearly seen. A fine needle was inserted into the vein and a total of 2 mL venous blood sample was collected from each patient. All blood samples were placed in tripotassium ethylene diamine tetraacetic acid anticoagulation tubes. All measurements were analyzed using a Pentra ES 60 (Horiba Medical, Montpellier, France) within 30 minutes after blood collection. White blood cell count, platelet count, neutrophil count, lymphocyte count (as well as band neutrophils and segmented neutrophils) were obtained directly from the blood analyzer, while NLR and PLR were obtained by dividing neutrophil count or platelet count by lymphocyte count, respectively. Similarly, during the entire cohort study, we used the Siemens Diagnostic Products Corporation Immulite® assay to measure hCG, with the reference value for normal serum hCG results below 5 IU/L.

## Outcome

The primary outcome evaluated was the occurrence of postmolar GTN. The secondary outcomes evaluated were the occurrence of medical complications, high preevacuation hCG levels and the presence of abnormalities in blood count parameters among patients with CHM, before molar evacuation.

## Variables

The following population variables were studied: age (in years), number of gestations, births and abortions of the patient.

Clinical and biochemical variables evaluated included: gestational age at diagnosis (in weeks), largest uterine diameter (in millimeters, evaluated by pelvic-transvaginal ultrasound), preevacuation hCG serum level (in IU/L), occurrence of medical symptoms at presentation (%): hemorrhage, enlarged uterus for gestational age (defined as a uterus measuring at least 4 cm more than expected for gestational age), theca lutein cysts larger than 6 cm (measured by pelvic-transvaginal ultrasound); early-onset pre-eclampsia (blood pressure of 140mmHg or greater for systolic pressure or 90mmHg or greater for diastolic pressure, on two occasions at least 4 hours apart in a previously normotensive patient, in the presence of proteinuria of 300mg or more in 24 hours or on the presence of headache, visual turbidity, abdominal pain or altered laboratory tests, such as platelets $<100,000/mm^3$, hepatic enzyme elevation more than double the basal level, serum creatinine $>1.1mg/dL$ or double the baseline, or pulmonary edema and visual or brain disorders such as headache, scotomas, or convulsions) and hyperemesis (presence of five or more episodes of vomiting per day, with or without metabolic alterations) [22].

Time to remission (in weeks) was defined as the period between the molar evacuation and the third hCG measurement of under 5 IU/L. We used the criteria established by Fédération Internationale de Gynécologie et d'Obstétrique (FIGO) and European Society for Medical Oncology for postmolar GTN diagnosis: rising (more than 10%) hCG levels for three consecutive weeks or plateaued for four weeks, a histological diagnosis of choriocarcinoma or metastases detected during post-molar follow-up, particularly in the lungs and pelvis [23, 24].

## Statistical analysis

We calculated a minimal sample size to detect mean differences in NLR between postmolar GTN and spontaneous remission among Brazilian women with CHM, considering a power of 80%, a significance level of 5% and a standard deviation in NLR of 1.93 for the control group and a standard deviation of 1.94 for the treatment group, and a minimum difference to detect of 0.40. Considering the calculation of the minimum number of patients to be included in this study, to ensure the power of its conclusions, we required a sample size of at least 736 subjects.

Categorical variables were described as absolute and relative frequencies, while continuous variables as medians and interquartile ranges. To compare proportions, a Chi-square test was used and the Mann-Whitney U test was used to compare continuous variables.

The optimal cut-points for leukogram variables were estimated by the Receiver Operating Characteristic (ROC) curve by minimizing the Euclidean distance between the curve and the point (0,1) in the ROC space. For each of the evaluated laboratory markers, their sensitivity (Se), specificity (Sp), positive (PPV) and negative predictive value (NPV) for the occurrence of the outcome of interest were calculated.

For outcomes of interest, crude and age/hCG-adjusted odds ratios (aOR) with 95% confidence intervals (CI) were calculated using the Wald test for logistic regression. The estimated

optimal cut-points were categorized and used as predictors in their respective logistic regression.

The analyses were carried out using SAS software, version 9.4 and the R software, version 4.1.3, was performed for graphics. All analyses considered a significance level of 0.05.

## Results

Fig 1 is a flow diagram summarizing the derivation of the study population. From January 2015 to December 2020, 1,685 patients with HM were followed at Rio de Janeiro GTD RC. Among these, 1,298 underwent uterine evacuation at the RC. After excluding cases of PHM (394 patients), twin molar pregnancy (13 patients), those who got pregnant (5 patients) or were lost to postmolar follow-up (59 patients), the final study population comprised 827 cases of CHM. Among these, 696 (84.15%) had spontaneous remission and 131 (15.85%) developed postmolar GTN.

The clinical characteristics of the study cohort are presented in Table 1. We observed that cases of CHM with advanced maternal age (33 versus 26 years, p<0.001), larger uterine diameter (110 versus 104mm, p = 0.042) and with hemorrhage (67.18% versus 41.67%, p<0.001) and enlarged uterus for gestational age (42.75% versus 28.59%, p = 0.001) as medical symptoms at presentation were associated with the development of postmolar GTN, and therefore longer time to remission (14 versus 9 weeks, p<0.001) when compared to patients who had spontaneous remission, respectively.

Table 2 shows that the difference in platelet count (257,500 versus 258,000mm$^3$, p = 0.803) and leukogram assessment, especially NLR (2.79 versus 2.64, p = 0.307) or PLR (129.26 versus 123.75, p = 0.610), were not associated with the occurrence of postmolar GTN, with the exception of the preevacuation hCG level (235.497 IU/L versus 154,409 IU/L, p<0.001) when compared to patients who had spontaneous remission, respectively.

To obtain the optimal cut-off among the different laboratory markers studied in the complete blood count, ROC curves for the occurrence of postmolar GTN were studied as shows in Fig 2. Table 3 present data, for each of these laboratory markers, including the area under the curve, its Se, Sp, PPV and NPV for the occurrence of the outcome evaluated. Platelet count and band neutrophils had the highest PPV (0.85 and 0.84), albeit with modest sensitivity (0.55 and 0.49) and specificity (0.49 and 0.52), respectively.

Using these optimal cut-offs from the ROC curves in multivariable logistic regression, adjusted for possible confounding variables of age and preevacuation hCG level, already known to be associated with the development of GTN, Table 4 shows that only the occurrence of ≥2 medical complications at presentation (aOR: 1.96, CI 95%: 1.29–2.98, p = 0.001) and preevacuation hCG ≥100,000 IU/L (aOR: 2.16, CI 95%: 1.32–3.52, p = 0.002) were able to predict the occurrence of postmolar GTN. None of the complete blood count parameters nor their ratios were prognostic.

## Discussion

While the occurrence of at least two medical complications at presentation and higher hCG preevacuation levels were associated with an increased chance of developing GTN during postmolar follow-up, none of the blood count parameters obtained before molar evacuation, notably platelet count, NLR or PLR, were able to predict the occurrence of postmolar GTN in the CHM population studied.

The intricate molecular mechanisms involved in the progression of CHM into GTN are not fully understood. It is noteworthy that our data reinforce the perception that the early diagnosis of CHM, while making the clinical presentation of this disease milder [25], does

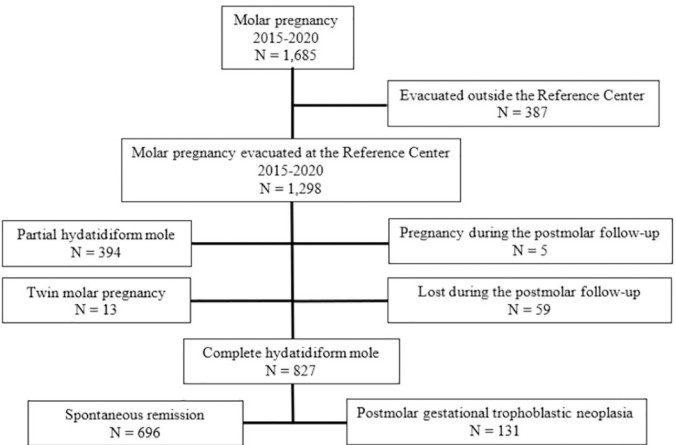

**Fig 1. Flow diagram summarizing the derivation of the study population.**

not have a protective effect against the occurrence of postmolar GTN [26]. This suggests that the propensity to develop GTN may already be present in the trophoblast tissue even before uterine evacuation. The presence of medical complications at presentation and the preevacuation hCG level have been classically associated with this outcome [1]. This study adds to the prior knowledge in the field by showing that the findings of the hemogram performed before molar evacuation, evaluated by optimal cut-off points, obtained by ROC curves, and adjusted for confounding variables, were not significantly associated with the development of postmolar GTN, as suggested by previous, smaller studies.

**Table 1. Clinical characteristics of patients with complete hydatidiform mole associated with spontaneous remission or progression to gestational trophoblastic neoplasia.**

| Variables | HM [a] with SR [b] (N = 696) | HM [a] with progression to GTN [c] (N = 131) | p-value [*] |
|---|---|---|---|
| Age (years) [#] | 26 (20–34) | 33 (24–41) | < 0.001 |
| Gestation [#] | 2 (1–3) | 2 (1–3) | 0.726 |
| Parity [#] | 0 (0–1) | 1 (0–1) | 0.236 |
| Abortion [#] | 0 (0–1) | 0 (0–1) | 0.605 |
| Gestational age at diagnosis (weeks) [#] | 11 (9–14) | 10 (9–13) | 0.099 |
| Largest uterine diameter (millimeter) [#] | 104 (82–123) | 110 (82–140) | 0.042 |
| Medical symptoms at presentation (%) | | | |
| *Hemorrhage* | 290 (41.67) | 88 (67.18) | < 0.001 |
| *Enlarged uterus for gestational age* | 199 (28.59) | 56 (42.75) | 0.001 |
| *Theca lutein cysts* | 37 (5.32) | 13 (9.92) | 0.042 |
| *Preeclampsia* | 23 (3.30) | 5 (3.82) | 0.766 |
| *Hyperemesis* | 157 (22.56) | 35 (26.72) | 0.301 |
| Time to remission (weeks) [#] | 9.00 (6.00–9.71) | 14.00 (10.00–19.00) | < 0.001 |

[#] Median and interquartile range.

[*] Chi-square or Mann-Whitney tests.

[a]. CHM. Complete hydatidiform mole.

[b]. SR. Spontaneous remission.

[c]. GTN. Gestational trophoblastic neoplasia.

**Table 2. Laboratory findings of patients with complete hydatidiform mole associated with spontaneous remission or progression to gestational trophoblastic neoplasia.**

| Variables [#] | HM [a] with SR [b] | HM [a] with progression to GTN [c] | p-value [*] |
|---|---|---|---|
| hCG [d] preevacuation (IU/L) | 154,409 (67,531–310,160) | 235,497 (110,034–671,275) | < 0.001 |
| Platelets (mm$^3$) | 258,000 (214,000–299,000) | 257,500 (212,000–303,000) | 0.803 |
| Leukogram | | | |
| *White blood count (mm$^3$)* | 7,478 (6,120–9,450) | 7,557 (6,080–9,476) | 0.862 |
| *Lymphocyte (mm$^3$)* | 2,046 (1,620–2,541) | 1,980 (1,550–2,520) | 0.535 |
| *Neutrophil (mm$^3$)* | 5,463 (4,148–7,052) | 5,613 (4,290–7,100) | 0.684 |
| *Band neutrophils (mm$^3$)* | 160 (82–292) | 156 (91–300) | 0.750 |
| *Segmented neutrophils (mm$^3$)* | 5,340 (4,080–6,825) | 5,373 (4,212–6,920) | 0.741 |
| Ratio neutrophil/lymphocyte | 2.64 (2.03–3.63) | 2.79 (2.19–3.61) | 0.307 |
| Ratio platelets/lymphocyte | 123.75 (97.81–161.80) | 129.26 (98.02–165.14) | 0.610 |

[#] Median and interquartile range.

[*] Mann-Whitney test.

[a]. HM. Hydatidiform mole.

[b]. SR. Spontaneous remission.

[c]. GTN. Gestational trophoblastic neoplasia.

[d]. hCG. Human chorionic gonadotropin.

Although most studies that evaluate the laboratory findings of the complete blood count compare patients with HM and a control formed by non-pregnant [27], healthy pregnancies [28] or even with cases of abortion [29, 30], three studies were identified that evaluated these findings among HM patients with spontaneous remission or that developed postmolar GTN. Guzel et al. found an association between NLR and the occurrence of postmolar GTN, but with values more than three times higher than we reported in the current study (8.96 versus 2.58, respectively) [18]. This may have occurred because these authors evaluated only 8 patients with postmolar GTN, a number 100 times lower than ours. The same limitation is present in the study by Yayla et al. who, when comparing 13 cases of postmolar GTN, found significantly lower levels in the PLR of patients who developed GTN [19]. Verit, reported that the platelet count was significant lower in patients who would develop postmolar GTN [20]. However, in addition to not comparing the platelets count obtained by the optimal cut-off point from ROC curves, the results of these three studies were not adjusted for confounding variables, in a multivariable logistic regression, as here. This is essential to exclude the influence of other variables on the results found. It is worth noting, for example, that, although discrete, the decrease in lymphocyte and platelet counts is expected with age, which, inversely, increases the occurrence of postmolar GTN, showing the importance of eliminating the effect of these confounders in this study.

Generally, cancer has many common features with infectious or inflammatory responses to trauma such as immune activation, acute phase response and systemic inflammation [16]. In this context, numerous studies have shown an association between high levels of NLR and tumor outcomes, making this marker frequently included in the prognostic evaluation of several solid tumors [31]. However, an extensive review of 204 meta-analyses of observational studies found that in only 60 meta-analyses (29%) an elevated NLR finding was, in fact, significantly associated with tumor prognosis. Furthermore, that paper also drew attention to the great heterogeneity of the studies evaluated, many with small sample sizes, generating small study effect biases, which need to be considered [32]. Another point is that many of the associations between blood count findings and oncologic prognosis come from observational studies

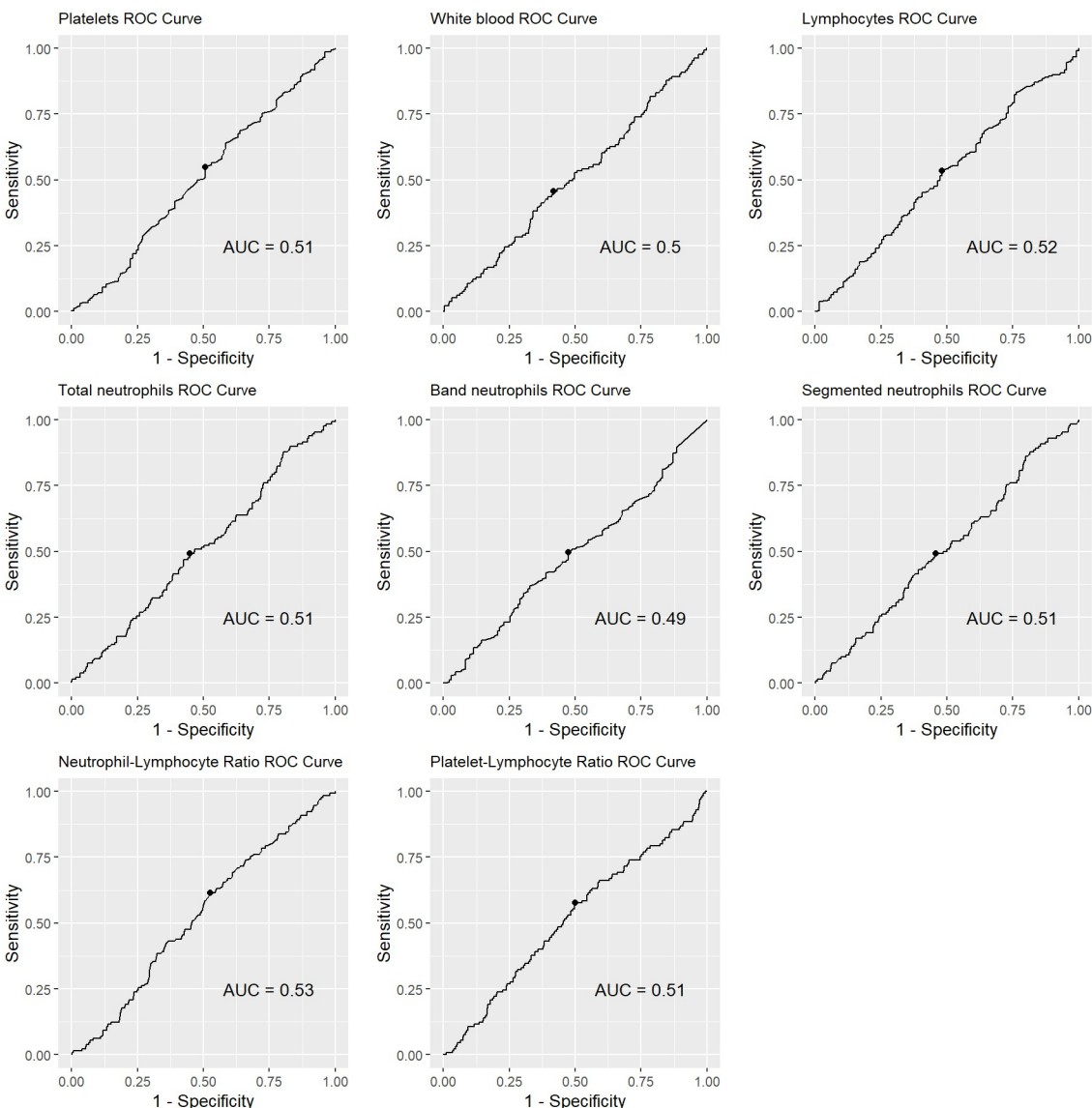

**Fig 2. Receiver operating characteristic curve demonstrating area under curve of white blood count, lymphocytes, platelets count, total neutrophils, band neutrophils, segmented neutrophils, neutrophil-lymphocyte ratio and platelet-lymphocyte ratio for the occurrence of gestational trophoblastic neoplasia among patients with complete hydatidiform mole.**

in which results that lie between OR 0.33 and 3 are considered significant, in which there is a risk of large bias potentially leading to spurious associations [33], demanding caution in the interpretation of these results.

Although the NLR and other blood cell component counts were unable to predict the cases of CHM that will develop postmolar GTN, it would be important to evaluate these markers in cases of postmolar GTN, in order to assess whether there is a relationship between these findings and the occurrence of chemoresistance, high risk disease, number of chemotherapy cycles to remission and time to remission, and survival rate or recurrence. Perhaps, studying established neoplasia (GTN), instead of pre-malignant disease (CHM) can provide better prognostic parameters. Although new modern technologies such as microRNA have shown promising

**Table 3. Predictive variables of the occurrence of gestational trophoblastic neoplasia among patients with complete hydatidiform mole obtained by receiver operating characteristic curves.**

| Variables | Area under curve | Optimal cut off | Se [a] | Sp [b] | PPV [c] | NPV [d] |
|---|---|---|---|---|---|---|
| Platelets (mm³) | 0.51 | 253,000 | 0.55 | 0.49 | 0.85 | 0.17 |
| Leukogram | | | | | | |
| White blood count (mm³) | 0.50 | 8,010 | 0.46 | 0.58 | 0.17 | 0.85 |
| Lymphocyte (mm³) | 0.52 | 1,998 | 0.53 | 0.52 | 0.85 | 0.17 |
| Neutrophil (mm³) | 0.51 | 5,720 | 0.49 | 0.55 | 0.17 | 0.85 |
| Band neutrophils (mm³) | 0.49 | 162 | 0.49 | 0.52 | 0.84 | 0.16 |
| Segmented neutrophils (mm³) | 0.51 | 5,544 | 0.49 | 0.54 | 0.17 | 0.85 |
| Neutrophil/lymphocyte ratio | 0.53 | 2.55 | 0.61 | 0.47 | 0.18 | 0.86 |
| Platelets/lymphocyte ratio | 0.51 | 123.75 | 0.57 | 0.50 | 0.18 | 0.86 |

[a]. Se. Sensitivity.

[b]. Sp. Specificity.

[c]. PPV. Predictive positive value.

[d]. NPV. Negative predictive value.

results in predicting the prognosis of CHM, their cost and availability limit their clinical use [13], which contrasts with NLR and other blood cell counts that are simple, inexpensive and widely available laboratory tests, which need to be further evaluated.

The greatest strength of our study was the use of an adequate statistical methodology that could identify the predictive factor of the outcome, excluding the effect of confounding variables. Furthermore, our study presented the largest series evaluating the potential relationship between the blood count in patients with hydatidiform mole in the literature, exceeding the minimum number of patients necessary to guarantee the power of its results. However, our

**Table 4. Logistic regression for prognostic factors associated with development of gestational trophoblastic neoplasia after complete hydatidiform mole.**

| Variables | cOR [a] * (95% CI) | p-value | aOR [b] * (95% CI) | p-value |
|---|---|---|---|---|
| Gestational age > 10 weeks | 0.72 (0.46–1.13) | 0.150 | 0.76 (0.46–1.25) | 0.275 |
| Occurrence ≥ 2 medical complications [#] | 2.22 (1.52–3.24) | <0.001 | 1.96 (1.29–2.98) | 0.001 |
| Preevacuation hCG [c,d] ≥100.000 IU/L | 2.14 (1.32–3.48) | 0.002 | 2.16 (1.32–3.52) | 0.002 |
| Platelets ≥ 253,000 mm³ | 0.85 (0.58–1.24) | 0.397 | 0.93 (0.62–1.41) | 0.742 |
| Leukogram | | | | |
| White blood count ≥ 8,010 mm³ | 1.18 (0.81–1.72) | 0.391 | 1.08 (0.71–1.64) | 0.710 |
| Lymphocyte ≥ 1,998 mm³ | 0.80 (0.55–1.17) | 0.251 | 0.89 (0.59–1.35) | 0.593 |
| Neutrophil ≥ 5,720 mm³ | 1.20 (0.82–1.74) | 0.349 | 1.08 (0.71–1.63) | 0.717 |
| Band neutrophils ≥ 162 mm³ | 0.91 (0.63–1.33) | 0.628 | 0.92 (0.61–1.39) | 0.686 |
| Segmented neutrophils ≥ 5,544 mm³ | 1.15 (0.79–1.67) | 0.477 | 1.03 (0.68–1.56) | 0.875 |
| Ratio neutrophil/lymphocyte ≥ 2.55 | 1.44 (0.98–2.12) | 0.060 | 1.29 (0.85–1.96) | 0.228 |
| Ratio platelets/lymphocyte ≥ 123.75 | 1.36 (0.93–1.99) | 0.108 | 1.49 (0.98–2.25) | 0.061 |

*. Wald test for logistic regression.

[#]. Medical complications: hemorrhage, enlarged uterus for gestational age, theca lutein cysts, preeclampsia or hyperemesis.

[a]. cOR. Crude odds ratio.

[b]. aOR. Adjusted odds ratio by age and hCG.

[c]. hCG. Human chorionic gonadotropin.

[d]. Adjusted odds ratio by age

study does have several limitations. The study is retrospective in nature and retrospective studies may potentially be prone to bias as records may be less complete and have less accurate record keeping. In addition, we only focused on the common parameters of the complete blood count and did not evaluate other hematologic variables such as red cell distribution width, mean corpuscular volume, or rare lymphocyte subsets.

In conclusion, even though the blood count results were not able to predict the progression of CHM into GTN, the occurrence of medical complications at presentation and higher pree-vacuation hCG levels were significantly associated with postmolar GTN and may be useful to guide individualized clinical decisions in post-molar follow-up and treatment of these patients. However, it would be helpful to identify an inexpensive and widely available laboratory prognostic markers to aid in the prediction of postmolar GTN.

## Author Contributions

**Conceptualization:** Antonio Braga, Ana Clara Canelas, Berenice Torres, Luana Giongo Pedrotti, Marina Bessel, Kevin M. Elias, Neil S. Horowitz, Ross S. Berkowitz.

**Data curation:** Antonio Braga, Ana Clara Canelas, Berenice Torres, Ana Paula Vieira dos Santos Esteves, Joffre Amim Junior, Jorge Rezende Filho.

**Formal analysis:** Antonio Braga, Marina Bessel, Kevin M. Elias, Neil S. Horowitz, Ross S. Berkowitz.

**Funding acquisition:** Kevin M. Elias, Neil S. Horowitz, Ross S. Berkowitz.

**Investigation:** Antonio Braga, Ana Clara Canelas, Ana Paula Vieira dos Santos Esteves, Joffre Amim Junior, Kevin M. Elias, Neil S. Horowitz, Ross S. Berkowitz.

**Methodology:** Antonio Braga, Ana Clara Canelas, Berenice Torres, Luana Giongo Pedrotti, Marina Bessel, Ana Paula Vieira dos Santos Esteves, Joffre Amim Junior, Jorge Rezende Filho, Kevin M. Elias, Neil S. Horowitz, Ross S. Berkowitz.

**Project administration:** Antonio Braga, Ana Clara Canelas, Berenice Torres, Ana Paula Vieira dos Santos Esteves, Jorge Rezende Filho.

**Supervision:** Antonio Braga, Joffre Amim Junior, Jorge Rezende Filho.

**Validation:** Antonio Braga, Luana Giongo Pedrotti, Marina Bessel, Jorge Rezende Filho.

**Visualization:** Joffre Amim Junior, Ross S. Berkowitz.

**Writing – original draft:** Antonio Braga, Izildinha Maesta, Luana Giongo Pedrotti, Marina Bessel, Joffre Amim Junior, Jorge Rezende Filho, Kevin M. Elias, Neil S. Horowitz, Ross S. Berkowitz.

**Writing – review & editing:** Antonio Braga, Izildinha Maesta, Luana Giongo Pedrotti, Marina Bessel, Joffre Amim Junior, Jorge Rezende Filho, Kevin M. Elias, Neil S. Horowitz, Ross S. Berkowitz.

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
