## [Decision Letter · Decision Letter 0]

10 Oct 2022

PONE-D-22-23500Neutrophil/lymphocyte ratio and other blood cell component counts are not associated with the development of postmolar gestational trophoblastic neoplasiaPLOS ONE

Dear Dr. Braga,

Thank you for submitting your manuscript to PLOS ONE. After careful consideration, we feel that it has merit but does not fully meet PLOS ONE’s publication criteria as it currently stands. Therefore, we invite you to submit a revised version of the manuscript that addresses the points raised during the review process.

We look forward to receiving your revised manuscript.

Kind regards,

Roberto Magalhães Saraiva, MD, PhD

Academic Editor

PLOS ONE

Journal Requirements:

Reviewers' comments:

Reviewer's Responses to Questions

**Comments to the Author**

1. Is the manuscript technically sound, and do the data support the conclusions?

Reviewer #1: Yes

2. Has the statistical analysis been performed appropriately and rigorously? 

Reviewer #1: Yes

3. Have the authors made all data underlying the findings in their manuscript fully available?

Reviewer #1: Yes

4. Is the manuscript presented in an intelligible fashion and written in standard English?

Reviewer #1: Yes

5. Review Comments to the Author

Reviewer #1: Many thanks for asking me to review this retrospective study from an internationally renowned GTD collaborative group. The study aimed to evaluate the relationship between neutrophil/lymphocyte ratios (NLR) and platelet/lymphocyte ratios and the occurrence of gestational trophoblastic neoplasia (GTN) after complete hydatidiform mole (CHM)

among Brazilian women. High pre-treatment NLR for example can be associated with poorer outcomes in cancer patients with variation in the size of this association across studies of cancer patients.

The paper is well written and clear and involves a larger number of CHM pts. The clear outcome, after adjusting for potential confounders, is that there is no association here with the development of post molar GTN from CHM. This is worthy of publication for the international GTN community as some much smaller series have reported a potential association.

I have the following minor comments:

Was there a set time frame before evacuation in which the blood count was taken or was it on the day of the evacuation - would this have any influence on the results

Are there any subgroups of patients e.g. ethic groups - where there may be stronger associations?

Can the authors expand on why they think that Neutrophil/lymphocyte ratio and other blood cell component counts are not predictive the development of postmolar gestational trophoblastic neoplasia from CHM. Is this the fact that CHM is a pre-malignant diagnosis, clearly different from e.g. NLR trying to predict outcome in established cancer? Would it be useful to look at a later stage e.g. NLR prior to treatment in GTN to see if this may have an association with outcome e.g. resistance, survival in high risk disease?

Perhaps the authors can add to the discussion at the end some further detail in how they see future development of predicting the development of post-molar GTN e.g. microRNA etc

6. PLOS authors have the option to publish the peer review history of their article (what does this mean?). If published, this will include your full peer review and any attached files.

Reviewer #1: No

---

## [Author Response · Author response to Decision Letter 0]

15 Oct 2022

Rio de Janeiro, Brazil

October, 2022.

Dr. Emily Chenette, PhD. Editor-in-Chief, PLOS ONE

Dear Dr. Emily Chenette and the members of the Editorial Board:

Response to reviewers

Paper: “Neutrophil/lymphocyte ratio and other blood cell component counts are not associated with the development of postmolar gestational trophoblastic neoplasia”

Initially, we would like to highlight that we made a surname correction, changing the original name from Berenice Nogueira to Berenice Torres. In addition, we linked a new institution to the first author: Postgraduate Program in Applied Health Sciences, Vassouras University. Rio de Janeiro – RJ, Brazil.

Editor review.

REPLY: We appreciate the comment and agree with the reviewer. Please note that the necessary corrections have been made.

2. Please provide additional details regarding participant consent. In the ethics statement in the Methods and online submission information, please ensure that you have specified (1) whether consent was informed and (2) what type you obtained (for instance, written or verbal, and if verbal, how it was documented and witnessed). If your study included minors, state whether you obtained consent from parents or guardians. If the need for consent was waived by the ethics committee, please include this information. If you are reporting a retrospective study of medical records or archived samples, please ensure that you have discussed whether all data were fully anonymized before you accessed them and/or whether the IRB or ethics committee waived the requirement for informed consent. If patients provided informed written consent to have data from their medical records used in research, please include this information.

REPLY: We appreciate the comment and agree with the reviewer. Please note that we already presented this declaration in the methods section, in the original submission, as follows: “The study was done with anonymized patient records, so the Ethics Committees waived the need for obtaining individual informed consent.”.

REPLY: We appreciate the comment and agree with the reviewer. We have included the following Data Availability Statement as follows: “Data Availability Statement. Due to ethical issues related to the public disclosure of a database of a disease that is not highly prevalent, coming from a delimited geographical area, with the potential to breach data confidentiality, even in the face of data de-identification, we emphasize that any data referring to the study: “Neutrophil/lymphocyte ratio and other blood cell component counts are not associated with the development of postmolar gestational trophoblastic neoplasia” must be requested directly from the Research Ethics Committee of the Maternity School of Rio de Janeiro Federal University, with due justification analyzed by this institutional review board, in order to preserve the interests of research participants, following compliance with research standards in Brazil (CNS/CONEP/CEP 466/2012).

The Research Ethics Committee of the Maternity School of Rio de Janeiro Federal University can be contacted directly through the institutional email: cep@me.ufrj.br or directly by phone: 55.21.2285-7935.”.

REPLY: We appreciate the comment and agree with the reviewer. We have made the necessary changes in this revised version.

REPLY: We appreciate the comment and agree with the reviewer. We emphasize that we do not include in our references any article that has been retracted and have made no changes to our reference list.

Reviewer 1. Many thanks for asking me to review this retrospective study from an internationally renowned GTD collaborative group. The study aimed to evaluate the relationship between neutrophil/lymphocyte ratios (NLR) and platelet/lymphocyte ratios and the occurrence of gestational trophoblastic neoplasia (GTN) after complete hydatidiform mole (CHM) among Brazilian women. High pre-treatment NLR for example can be associated with poorer outcomes in cancer patients with variation in the size of this association across studies of cancer patients. The paper is well written and clear and involves a larger number of CHM pts. The clear outcome, after adjusting for potential confounders, is that there is no association here with the development of post molar GTN from CHM. This is worthy of publication for the international GTN community as some much smaller series have reported a potential association. I have the following minor comments:

1. Was there a set time frame before evacuation in which the blood count was taken or was it on the day of the evacuation - would this have any influence on the results.

REPLY: We appreciate the comment and agree with the reviewer. The issue is relevant and we added the following sentence in the description of the methods: "In all patients, the blood test was collected within 6 hours before surgery".

2. Are there any subgroups of patients e.g. ethic groups - where there may be stronger associations?

REPLY: We appreciate the opportunity to clarify these aspects. Although previous studies have attributed different outcomes of hydatidiform mole, in relation to certain ethnic population groups, no Brazilian study has been carried out in this sense. This is due to the Brazilian ethnic background being extremely mixed, making it difficult to ethnically characterize patients. Furthermore, Brazilian legislation prohibits physicians from establishing the ethnic classification of their patients, which in Brazil is done according to self-declaration, which, in itself, makes it difficult to analyze this variable.

3. Can the authors expand on why they think that Neutrophil/lymphocyte ratio and other blood cell component counts are not predictive the development of postmolar gestational trophoblastic neoplasia from CHM. Is this the fact that CHM is a pre-malignant diagnosis, clearly different from e.g. NLR trying to predict outcome in established cancer? Would it be useful to look at a later stage e.g. NLR prior to treatment in GTN to see if this may have an association with outcome e.g. resistance, survival in high risk disease?

REPLY: We appreciate the comment and agree with the reviewer. We have added the following sentences in the Discussion section to contextualize the reviewer's comment, as follows: “Although the NLR and other blood cell component counts were unable to predict the cases of CHM that will develop postmolar GTN, it would be important to evaluate these markers in cases of postmolar GTN, in order to assess whether there is a relationship between these findings and the occurrence of chemoresistance, high risk disease, number of chemotherapy cycles to remission and time to remission, and survival rate or recurrence. Perhaps, studying established neoplasia (GTN), instead of pre-malignant disease (CHM) can provide better prognostic parameters.”.

4. Perhaps the authors can add to the discussion at the end some further detail in how they see future development of predicting the development of post-molar GTN e.g. microRNA etc.

REPLY: We appreciate the comment and agree with the reviewer. We have added the following sentences in the Discussion section to contextualize the reviewer's comment, as follows: “Although new modern technologies such as microRNA have shown promising results in predicting the prognosis of CHM, their cost and availability limit their clinical use [34], which contrasts with NLR and other blood cell counts that are simple, inexpensive and widely available laboratory tests, which need to be further evaluated.”.

34. Lin LH, Maestá I, St Laurent JD, Hasselblatt KT, Horowitz NS, Goldstein DP, Quade BJ, et al. Distinct microRNA profiles for complete hydatidiform moles at risk of malignant progression. Am J Obstet Gynecol. 2021;224(4):372.e1-372.e30. doi: 10.1016/j.ajog.2020.09.048. 

Sincerely,

Antonio Braga MD and Ross Berkowitz MD

For the authors

---

## [Decision Letter · Decision Letter 1]

6 Nov 2022

Neutrophil/lymphocyte ratio and other blood cell component counts are not associated with the development of postmolar gestational trophoblastic neoplasia

PONE-D-22-23500R1

Dear Dr. Braga,

We’re pleased to inform you that your manuscript has been judged scientifically suitable for publication and will be formally accepted for publication once it meets all outstanding technical requirements.

Kind regards,

Roberto Magalhães Saraiva, MD, PhD

Academic Editor

PLOS ONE

Additional Editor Comments (optional):

Reviewers' comments:

Reviewer's Responses to Questions

**Comments to the Author**

1. If the authors have adequately addressed your comments raised in a previous round of review and you feel that this manuscript is now acceptable for publication, you may indicate that here to bypass the “Comments to the Author” section, enter your conflict of interest statement in the “Confidential to Editor” section, and submit your "Accept" recommendation.

Reviewer #1: All comments have been addressed

2. Is the manuscript technically sound, and do the data support the conclusions?

Reviewer #1: Yes

3. Has the statistical analysis been performed appropriately and rigorously? 

Reviewer #1: Yes

4. Have the authors made all data underlying the findings in their manuscript fully available?

Reviewer #1: Yes

5. Is the manuscript presented in an intelligible fashion and written in standard English?

Reviewer #1: Yes

6. Review Comments to the Author

Reviewer #1: The authors have responded with appropriate answers and they have made appropriate edits made to the manuscript

7. PLOS authors have the option to publish the peer review history of their article (what does this mean?). If published, this will include your full peer review and any attached files.

Reviewer #1: No

---

## [Editor Report · Acceptance letter]

9 Nov 2022

PONE-D-22-23500R1 

Neutrophil/lymphocyte ratio and other blood cell component counts are not associated with the development of postmolar gestational trophoblastic neoplasia 

Dear Dr. Braga:

I'm pleased to inform you that your manuscript has been deemed suitable for publication in PLOS ONE. Congratulations! Your manuscript is now with our production department. 

Kind regards, 

on behalf of

Dr. Roberto Magalhães Saraiva 

Academic Editor

PLOS ONE